# Optical Turbulence Characteristics in the Upper Troposphere–Lower Stratosphere over the Lhasa within the Asian Summer Monsoon Anticyclone

**Kun Zhang** [1,2], **Feifei Wang** [1,2], **Ningquan Weng** [1,2], **Xiaoqing Wu** [1,2], **Xuebin Li** [1,2] **and Tao Luo** [1,2,*]

1    Key Laboratory of Atmospheric Optics, Anhui Institute of Optics and Fine Mechanics, Hefei Institutes of Physical Science, Chinese Academy of Sciences, Hefei 230031, China
2    Advanced Laser Technology Laboratory of Anhui Province, Hefei 230037, China
*    Correspondence: luotao@aiofm.ac.cn

**Abstract:** The high elevation, complex topography, and unique atmospheric circulations of the Tibetan Plateau (TP) make its optical turbulence characteristics different from those in low-elevation regions. In this study, the characteristics of the atmospheric refractive index structure constant ($C_n^2$) profiles in the Lhasa area at different strength states of the Asian summer monsoon anticyclone (ASMA) are analyzed based on precious in situ sounding data measured over the Lhasa in August 2018. $C_n^2$ in the upper troposphere–lower stratosphere fluctuates significantly within a few days during the ASMA, particularly in the upper troposphere. The effect of the ASMA on $C_n^2$ varies among the upper troposphere, tropopause, and lower stratosphere. The stronger and closer the ASMA is to Lhasa, the more pronounced is the "upper highs and lower lows" pressure field structure, which is beneficial for decreasing the potential temperature lapse rate. The decrease in static stability is an important condition for developing optical turbulence, elevating the tropopause height, and reducing the tropopause temperature. However, if strong high-pressure activity occurs at the lower pressure layer, such as at 500 hPa, an "upper highs and lower highs" pressure field structure forms over the Lhasa, increasing the potential temperature lapse rate and suppressing the convective intensity. Being almost unaffected by low-level atmospheric high-pressure activities, the ASMA, as the main influencing factor, mainly inhibits $C_n^2$ in the tropopause and lower stratosphere. The variations of turbulence intensity in UTLS caused by ASMA activities also have a great influence on astronomical parameters, which will have certain guiding significance for astronomical site testing and observations.

**Keywords:** Tibetan Plateau; optical turbulence; Asian summer monsoon anticyclone; upper troposphere-lower stratosphere

## 1. Introduction

The unique features of the Tibetan Plateau (TP), such as its complex terrain formed by high mountains and valleys, dramatic changes in the atmospheric environment, differences in atmospheric composition, and unique geographical climate and circulation characteristics, form different atmospheric optical turbulence characteristics over the TP from those of low-elevation plain regions [1]. The TP is the major energy source providing sensible and latent heat fluxes to the atmosphere depending on the turbulence processes that occur during land–atmosphere interactions for mass and energy exchanges [2]. The combined effect of the complex terrain of the TP and the heat source enables the development of turbulence in the middle and upper atmosphere over the region.

Strong Asian summer monsoon circulations exist above the TP, including deep convective activities and planetary-scale anticyclones, such as the South Asian high, SAH (hereinafter referred to as the Asian summer monsoon anticyclone (ASMA)) [3,4]. The ASMA is stable and strong in the vertical direction at 70–300 hPa and occupies almost

the entire upper troposphere–lower stratosphere (UTLS) area [5], which is closely related to plateau land-atmosphere heat transfer [6,7]. The ASMA is closely related to frequent convective activities, particularly during the period from June to September. The geographic location of the ASMA center varies over periods of a few days or even over longer periods [8]. The coupling of atmospheric circulation and convection that prevails over the TP during the summer results in the frequent occurrence of convective activities in the lower atmosphere. Less details are known about the influence of the ASMA on the thermodynamic structure of the atmosphere in the stratosphere.

The strong convective activities and ASMA on the TP affect the atmospheric components and their distribution in the UTLS of the Asian monsoon region by uplifting the lower atmosphere [9–11]. A turbulent atmosphere is an important transport medium in stratosphere–troposphere exchange (STE). First, convective injections can impact air and aerosol transport from the atmospheric boundary layer (ABL) to the UTLS [12,13]. In contrast, deep convection activities carry low concentrations of ozone and high concentrations of water vapor into the ASMA, which remain inside the ASMA for a period of time, and they are relatively isolated from the outside air and subsequently uploaded to the UTLS [14,15]. Moreover, the vertical distribution of atmospheric turbulence is one of the factors that must be inevitably considered in the astronomical site testing of ground-based astronomical optical telescopes [16]. Atmospheric turbulence is the main reason for the serious degradation of optical imaging quality, and it is also an important indicator for comparing the quality of astronomical sites [17,18].

TP has garnered significant attention as the third pole of the Earth. In the past three decades, researchers have conducted several atmospheric scientific experiments on the TP combined with numerical simulations. Some studies have been conducted mainly on the structure of the ABL over the TP and its surrounding areas, including studies on the occurrence and development of weather systems and the impact of the TP on atmospheric circulation [19,20].

However, the relationship between the optical turbulent structure and meteorological parameters in the UTLS has rarely been studied [21]. The results of observations and numerical simulations obtained in recent years indicate that the ASMA region surrounding Lhasa as the core area is an extremely important area, through which surface pollutants enter the global stratosphere. The transport of these materials into the stratosphere through atmospheric turbulence has important effects on the global climate and environment through microphysical, chemical, and radiative processes [22–24].

At present, most research on atmospheric turbulence structure in UTLS over the TP are studied based on model simulations or reanalysis, because in situ observed turbulent data are scarce. In consequence, details regarding the structure of atmospheric turbulence in the UTLS, STE process, and characteristics of atmospheric composition budgets are still lacking. The widely used measurement techniques of high-altitude atmospheric turbulence characteristics are roughly divided into two categories. One is the path-averaged turbulence intensity measurement technique, such as the optical triangulation method [16,25] and extension technology [26]; the other is the localized turbulence intensity measurement technique, such as the micro-thermal pulsation method. In situ turbulent measurement based on radiosonde is a simple but very reliable, effective, high-precision, and commonly used atmospheric detection method, particularly the micro-thermal sensors mounted on the balloon, which realizes the measurement of atmospheric turbulence in UTLS [27–30]. As such, not only basic atmospheric parameters such as temperature and humidity can be measured, but also the value of atmospheric refractive index structure constant $C_n^2$ in the middle and upper atmosphere is obtained simultaneously, which is an important parameter to measure turbulence intensity. This study analyzes the reasons for the large short-term fluctuations of $C_n^2$ in the Lhasa region from the aspects of atmospheric turbulence parameters, the ASMA, high-pressure activities at 500 hPa, and atmospheric circulation.

## 2. Methods

### 2.1. Sounding Data

From 3 to 18 August 2018, a thermal turbulent sounding experiment was conducted at the Lhasa Meteorological Bureau (91°06′E, 29°36′N) ("pentagram" in Figure 1) by the Hefei Institutes of Physical Science, Chinese Academy of Sciences. This experiment collected very precious high vertical resolution thermal turbulence sounding data, which provided reliable basis for the study of the fine atmospheric structure in UTLS and the verification of model simulation over the TP [30]. The sounding balloons were equipped with conventional meteorological sensors to measure the atmospheric temperature ($T$), humidity, pressure ($P$), and wind speed, along with two-channels turbulent meteorological radiosondes developed by the Anhui Institute of Optics and Fine Mechanics.

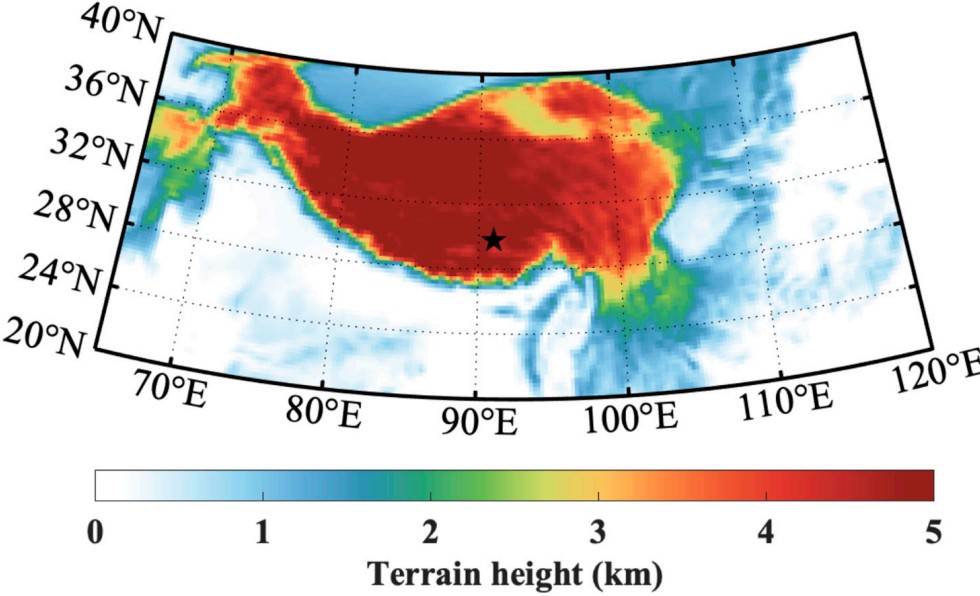

**Figure 1.** The elevation height of TP (color shaded) and the geographical location of Lhasa (pentagram). This figure was plotted using the Lambert conformal conic projection.

Each thermal turbulence radiosonde comprises two platinum wire probes (15 μm in diameter) separated by a distance of $r$ (=1 m). The platinum wire probes have linear resistance–temperature coefficients. The thermal turbulence radiosondes measure the temperature difference between the distance r and voltage change between the two microthermal probes [28–30]. Then, the temperature structure constant ($C_T^2$) in the inertial subrange can be calculated using the following equation [31]:

$$\left\langle \left[ T\left( \vec{x} \right) - T\left( \vec{x} + \vec{r} \right) \right]^2 \right\rangle = C_T^2 r^{\frac{2}{3}} \ (l_0 \ll r \ll L_0) \tag{1}$$

where $T\left( \vec{x} \right)$ and $T\left( \vec{x} + \vec{r} \right)$ denote the temperatures at two points, $\langle \cdots \rangle$ represents the ensemble average, and $l_0$ and $L_0$ represent the inner and outer scales of the turbulence, respectively (units of m).

$C_n^2$, the degree of refractive index fluctuation due to variations in atmospheric temperature and density [18,32], can be calculated by inputting temperature ($T$) and pressure ($P$) profiles, according to the relationship between $C_n^2$ and $C_T^2$ (Equation (2)):

$$C_n^2 = \left( 79 \times 10^{-6} \frac{P}{T^2} \right)^2 C_T^2 \tag{2}$$

The range of the response frequency of the thermal turbulent radiosonde is 0.1–30 Hz, and the minimum measurable standard deviation of the temperature fluctuation does not exceed 0.002 °C. In addition, the vertical resolution of the radiosondes was 30 m.

Five thermal turbulence radiosondes were launched during the experiments at about 19:30 local time (LT). The detailed experimental records are summarized in Table 1. Owing to weather and transmission interference problems, four valid data sets were obtained over 20 km above sea level (ASL, the height below refers specifically to ASL except for special explanations) in height.

**Table 1.** Detailed records of radiosonde experiments over the Lhasa.

| Radiosonde Number | Release Time (LT) | Release Altitude (m ASL) | Termination Altitude (m ASL) | Remark |
|---|---|---|---|---|
| 1 | 12 August 2018 19:24:36 | 3653.6 | 21,810.0 | Cloudy |
| 2 | 13 August 2018 19:28:03 | 3654.1 | 23,221.1 | Cloudy |
| 3 | 14 August 2018 19:40:46 | 3652.8 | 30,658.8 | Cloudy |
| 4 | 15 August 2018 19:04:16 | 3658.2 | 29,956.9 | Storm; micro-thermal sensors was destroyed |
| 5 | 16 August 2018 19:20:15 | 3652.2 | 31,206.0 | Cloudy |

### 2.2. $C_n^2$ Integrated Parameters

$C_n^2$ integrated parameters (the Fried parameter $r_0$, seeing $\varepsilon$, isoplanatic angle $\theta_0$) are of importance evaluation criteria for the astronomical site testing and the design of adaptive optics systems, defining as:

$$r_0 = \left[ 0.423 \left( \frac{2\pi}{\lambda} \right)^2 \sec\beta \int_0^\infty C_n^2(h)dh \right]^{-\frac{3}{5}} \tag{3}$$

$$\varepsilon = 5.25\lambda^{-\frac{1}{5}} \left[ \int_0^\infty C_n^2(h)dh \right]^{-\frac{3}{5}} = 0.98\frac{\lambda}{r_0} \tag{4}$$

$$\theta_0 = 0.057\lambda^{\frac{6}{5}} \left[ \int_0^\infty C_n^2(h)h^{\frac{5}{3}}dh \right]^{-\frac{3}{5}} \tag{5}$$

where, $\lambda$ (=550 nm for this study) is a given wavelength for visible light, $\beta$ is the zenith angle.

### 3. Results

### 3.1. ASMA Activities during the Experimental Period

SAH has significant multicenter characteristics [33], particularly bimodality [5,34], which is attributed to the warm preference of the SAH. Only the Tibetan mode is considered in this study. Figure 2 shows the distributions of the ECMWF geopotential height at 200-hPa pressure level on 12 August (Figure 2a), 13 (Figure 2b), 14 (Figure 2c), and 16 (Figure 2d), overlaid with the wind fields. The black square in Figure 2 denotes the location of Lhasa, whereas the black circle denotes the center of the SAH that is associated with the largest geopotential height. As shown in Figure 3a, the center of the SAH gradually moved northeastward during the experimental period. After 14 August, two SAH centers formed, with one located over the TP and the other over the Iranian Plateau.

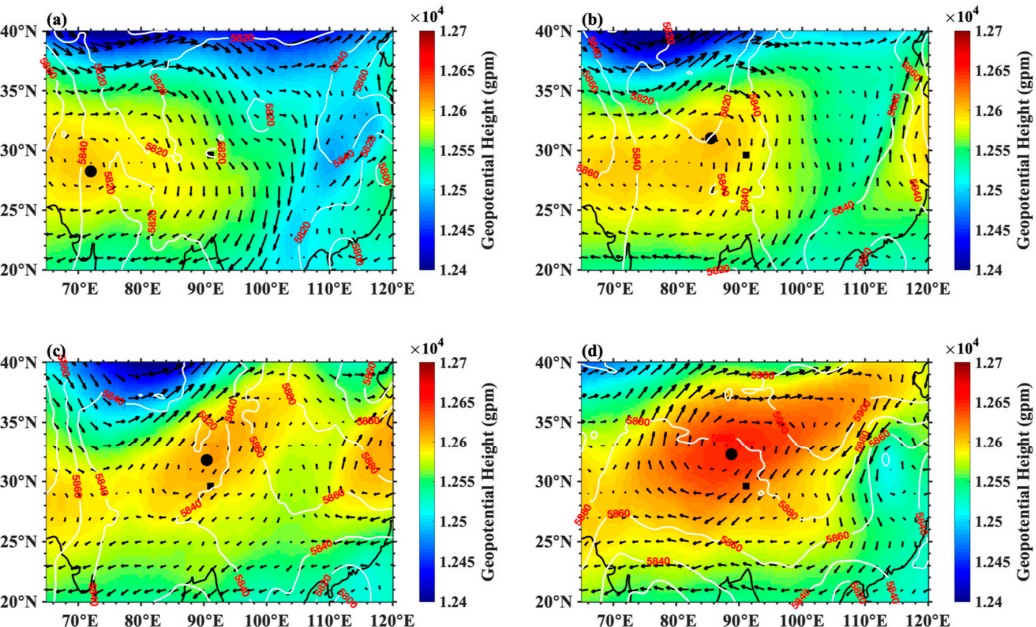

**Figure 2.** Distribution of the 200 hPa geopotential height (shaded), wind (vector), the 500 hPa geopotential height (contour), and the central region of the ASMA (the black square represents the geographic location of Lhasa, and the black dots represent the strongest negative vortex region. The same symbols are used below). The weather conditions during experimental periods were analyzed using the ERA-interim reanalysis data (http://www.ecmwf.int/, accessed on 17 January 2019) of the European Centre for Medium-Range Weather Forecasts (ECMWF). The horizontal resolutions of the meridional wind and relative vorticity were 2.5° × 2.5° and 0.25° × 0.25°, respectively. (**a**) 12 August 2018; (**b**) 13 August 2018; (**c**) 14 August 2018; and (**d**) 16 August 2018.

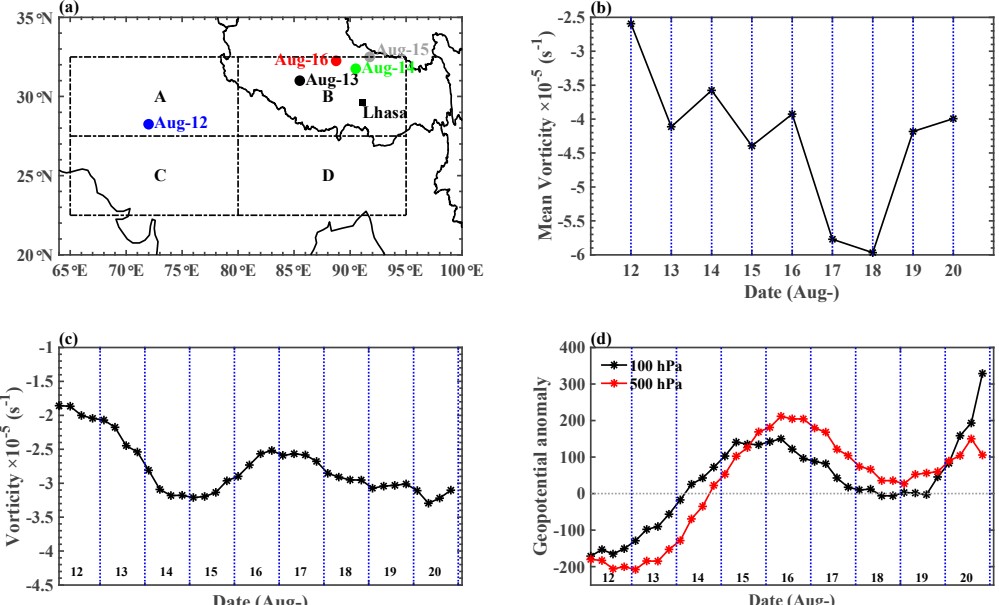

**Figure 3.** (**a**) The geographic locations of the ASMA centers from 12–16 August 2018. A-area (27.5–32.5°N, 65–80°E), B-area (27.5–32.5°N, 80–95°E), C-area (22.5–27.5°N, 65–80°E), and D-area (22.5–27.5°N, 80–95°E) were divided to identify the ASMA center (the black square represents the geographic location of Lhasa; the same symbols are used below); (**b**) the mean vorticity of 20 × 20 grids with the ASMA center considered as the geometric center, representing the strength of the ASMA; (**c**) the vorticity at 200 hPa over Lhasa from 12–20 August 2018; and (**d**) the geopotential height anomalies at 500 hPa and 100 hPa over Lhasa from 12–20 August 2018.

Figure 3 shows the development of the ASMA and its influence on Lhasa during the experimental period. The geolocation of the strongest ASMA center (dots in Figure 3a) is defined as the position of the greatest potential height in the strongest anticyclone area among the four areas (denoted as the A-area (27.5–32.5°N, 65–80°E), B-area (27.5–32.5°N, 80–95°E), C-area (22.5–27.5°N, 65–80°E), and D-area (22.5–27.5°N, 80–95°E)), according to the literatures [35,36]. In the physical sense, the criterion used to measure the strength of the ASMA is that the potential vorticity (PV) at the center of the anticyclone is smaller than that in the surrounding region [15,37,38]. The intensity of ASMA is shown in Figure 3b. The strength of the high-pressure system over the TP gradually increased with increasing geopotential height. Along with the development of a high-pressure system, the ASMA center moved to the TP after 12 August (onset), and its intensity was generally enhanced (13–15 August, early stage formation). After 16 August, the ASMA was fully established over the TP, with its intensity reaching the highest value.

The variations in relative vorticity at 200 hPa (Figure 3c) and geopotential height anomalies at 500 and 100 hPa over Lhasa (Figure 3d) are also presented. Along with the approach of the ASMA center toward Lhasa, the strength of the PV at 200 hPa over Lhasa increased, reached a maximum on 14 August, and thereafter gradually decreased. The anomalies at 100 hPa over Lhasa (Figure 3d) gradually increased from 12–15 August 2018, indicating that the transition from low to high pressure activity gradually occurred. Although the high-pressure activity at 500 hPa started to form with a one-day lag, it showed a more rapid growth trend and lasted longer. It can also be seen from the 500 hPa geopotential height field (Figure 2) that the high-pressure activity at 500 hPa gradually increased from the 13th, enveloping the entire TP on 16th. Notably, the geopotential height anomaly changed from negative to positive on 14 August, and the subsequent growing tendency was fiercer, which indicated that the high-pressure activity at 500 hPa became abnormally strong from 14 August.

### 3.2. Characteristics of $C_n^2$ under Different ASMA Strength States

We selected four representative profiles obtained at night on 12–14 and 16 August 2018, as shown in Figure 4, to analyze the variation characteristics of $C_n^2$ under different ASMA strength states over the Lhasa. There were strong maximum peaks of $C_n^2$ at 17–18 km (about 100 hPa). Certainly, a weak and thin maximum peak of Cn2 appeared at around 12 km (about 200 hPa), such as the 16 August 2018.

$C_n^2$ was largest on 13 August and smallest on 16 August. On 12 August, ASMA had a subtle impact on the $C_n^2$ profile over Lhasa. As the ASMA center approached Lhasa and its intensity increased, $C_n^2$ increased correspondingly on 13 and 14 August. Although these two days are in the middle stage of the ASMA, $C_n^2$ decreased on 14 August to a lower value than that recorded on 13 August in the range of ~15–20 km. With the departure and attenuated intensity of the ASMA, $C_n^2$ decreased rapidly on 16 August 2018.

In general, when the ASMA intensity was higher or the ASMA center was closer to Lhasa, a more pronounced "upper highs and lower lows" pressure field structure appeared over Lhasa. The stronger the convective activity, the greater the value of $C_n^2$. However, changes in the low-level pressure field, such as at 500 hPa, may have had a crucial impact on the vertical profile of $C_n^2$. A turning point occurred on 14 August, when the geopotential anomaly value changed from negative to positive, that is, the pressure field constructed in the UTLS changed from "upper highs and lower lows" to "upper highs and lower highs".

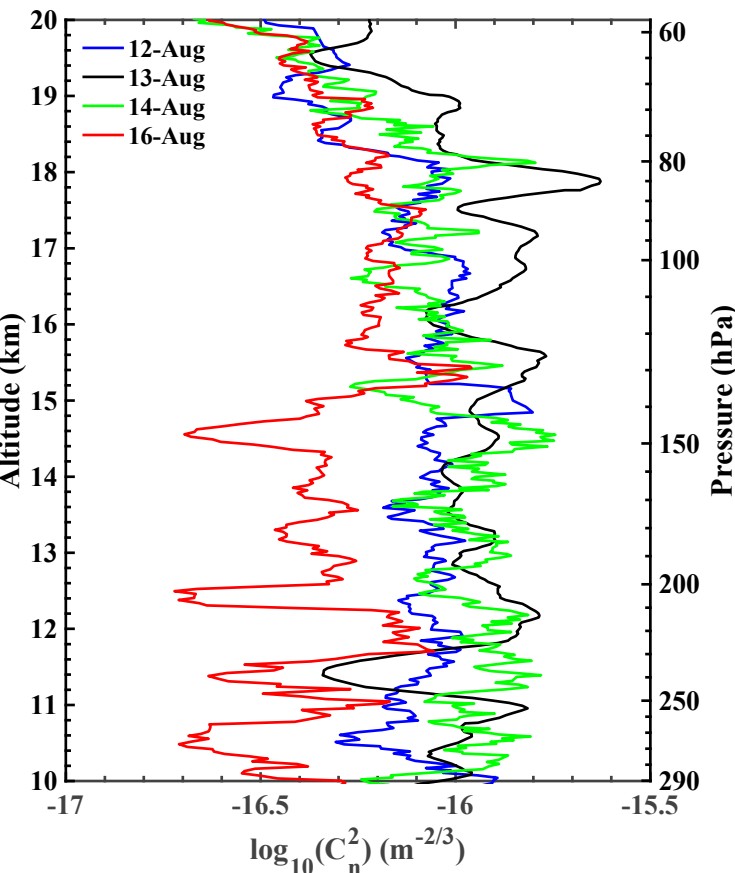

**Figure 4.** Vertical profiles of the atmospheric refractive index structure constant $C_n^2$ in the UTLS.

In comparison with that recorded on 16 August 2018, the high-pressure activity at the 500 hPa layer was weaker on 14 August, and the upward movement of the atmosphere was only slightly suppressed [37]. Therefore, $C_n^2$ on 14 August was higher than that on 16 August, but lower than that on 13 August.

### 3.3. Contribution of Atmospheric Turbulence in UTLS to the Total Integrated Parameters

The turbulent energy ratio (TER) in the range of 10–20 km, describing the contribution of atmospheric turbulence in this layer to $\varepsilon$ and $\theta_0$ of the total layer, can be calculated using the following equation [29]:

$$\text{TER}_\varepsilon = \frac{\varepsilon(i)^{5/3}}{\varepsilon(\text{total})^{5/3}} \times 100\% \tag{6}$$

$$\text{TER}_{\theta_0} = \frac{\theta_0(i)^{5/3}}{\theta_0(\text{total})^{5/3}} \times 100\% \tag{7}$$

where, $i$ stands for 10–20 km turbulent layer.

Table 2 summarizes the integrated contribution of seeing ($\varepsilon$) and isoplanatic angle ($\theta_0$) from the range of 10–20 km and the total integrated parameters. The atmospheric turbulence in the range of 10–20 km has a more significant proportion of $\varepsilon(\text{total})$ (more than 60%) and $\theta_0(\text{total})$ (more than 70%) over the Lhasa, which is consistent with the results of Gaomeigu site, Yunnan observatories, Chinese Academy of Science [29]. The $\varepsilon(i)$ ($\theta_0(i)$) differs 0.5″ (0.21″) between 13 and 16 August 2018, and $\text{TER}_\varepsilon$, as well as $\text{TER}_{\theta_0}$, varies so widely (more than 10%), which indicates that the variations of $C_n^2$ under different ASMA strength states are related with the astronomical observations.

**Table 2.** The contribution of seeing ($\varepsilon$ ) and isoplanatic angle ($\theta_0$ ) from the range of 10–20 km to total height layer.

|  |  | 12 August 2018 | 13 August 2018 | 14 August 2018 | 16 August 2018 |
|---|---|---|---|---|---|
| $\varepsilon$ ($''$) | $\varepsilon(i)$ | 1.08 | 1.32 | 1.19 | 0.82 |
|  | $\varepsilon(\text{total})$ | 1.30 | 1.63 | 1.60 | 1.11 |
|  | $\text{TER}_\varepsilon$ | 73.73% | 70.21% | 61.09% | 59.76% |
| $\theta_0$ ($''$) | $\theta_0(i)$ | 0.44 | 0.35 | 0.41 | 0.56 |
|  | $\theta_0(\text{total})$ | 0.40 | 0.30 | 0.35 | 0.45 |
|  | $\text{TER}_{\theta_0}$ | 85.58% | 78.82% | 77.84% | 69.65% |

## 4. Discussion

### 4.1. Potential Temperature Gradient

Equation (2) indicates that the most critical step in the parameterization scheme of atmospheric turbulence is how to parameterize $C_T^2$, and $C_n^2$ can be calculated logically. According to the dimensional analysis, Tatarski [31] defined the atmospheric temperature structure constant as follows:

$$C_T^2 = 1.6\varepsilon_\theta\varepsilon^{-\frac{1}{3}} \tag{8}$$

where $\varepsilon_\theta$ denotes the molecular diffusivity caused by temperature difference, and $\varepsilon$ is the turbulent energy dissipation rate. The energy of atmospheric turbulence mainly originates from the dynamic and thermal effects. The former implies that when there is wind direction and wind speed shear, the turbulent shear stress works on the air micro-clusters, whereas the latter implies that in an unstable atmosphere, the buoyant force works on the air micro-clusters that move vertically to increase the turbulence.

Bougeault and Lacarrere [39] parameterized $C_T^2$ as follows:

$$C_T^2 = 0.59L^{4/3}\left(\frac{\delta\bar{\theta}}{\delta z}\right)^2\varnothing_3 \tag{9}$$

where $L$ denotes the Bougeault–Lacarrere mixing length, $\varnothing_3$ is the reversed turbulent Prandtl number, and $\theta$ is the potential temperature. Refer to the detailed derivation process in the literature [40].

$$L = \sqrt{\frac{2e}{\frac{g}{\theta}\frac{\delta\bar{\theta}}{\delta z}}} \tag{10}$$

where $e$ denotes the turbulence energy. Parametric Equations (8)–(10) clearly indicate that, the potential temperature gradient is directly related to the buoyancy frequency, turbulent energy dissipation rate, and temperature structure constant. This is an indispensable and important parameter in the parameterization of atmospheric turbulence [41]. Therefore, under different ASMA intensities, the numerical changes in the potential temperature gradient can also reflect the thermal convection intensity.

### 4.2. Discussion on the Temperature Structure in UTLS

Radiosonde data were used to compare and analyze the atmospheric temperature structure from 12 to 16 August 2018. The temperatures above ~16 km fluctuated significantly during these days, where was stably controlled by ASMA. As shown in Figure 5b, the cold-point tropopause (CPT), which corresponds to the coldest temperature, was higher in the middle of the ASMA than in other stages, and this timing may have coincided with a decrease in static stability [42] and may be related to strong convective activities [43,44].

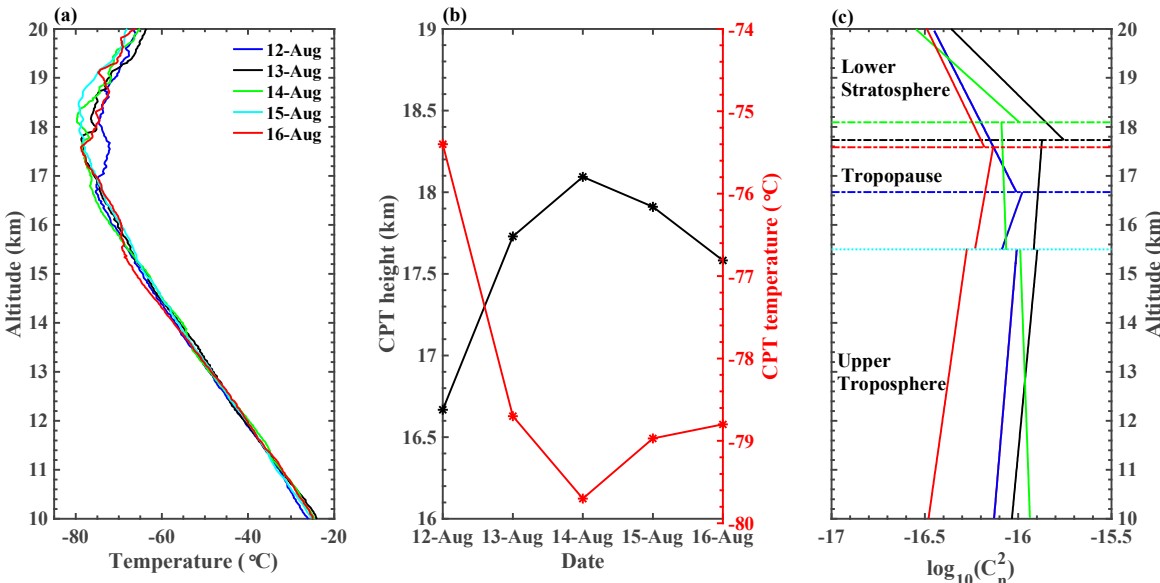

**Figure 5.** (**a**) Vertical temperature profiles from 12 to 16 August 2018. (**b**) Cold-point tropopause height (the black line) and temperature (the red line) from 12 to 16 August 2018. The lowest layer of the multi-tropopause serves as the CPT height. (**c**) First-order linear piecewise fitting to $\log_{10}(C_n^2)$ within the upper troposphere, tropopause, and lower stratosphere.

Table 3 summarizes the fitting results of $\log_{10}(C_n^2)$ using first-order linear fitting within the upper troposphere, tropopause, and lower stratosphere (Figure 5c), and the coefficients represent the increase rate of $\log_{10}(C_n^2)$, reaching respective maximum value at tropopause layer. However, when the ASMA intensity is large, the $\log_{10}(C_n^2)$ increase rate in the tropopause is weakened, and the $\log_{10}(C_n^2)$ in the lower stratosphere decreases.

**Table 3.** Increase rate of $\log_{10} C_n^2$ ($m^{7/3}$) within upper troposphere, tropopause, and lower stratosphere.

|  | 12 August 2018 | 13 August 2018 | 14 August 2018 | 16 August 2018 |
|---|---|---|---|---|
| Upper troposphere (10–15.5 km) | 0.02 | 0.02 | −0.01 | 0.04 |
| Tropopause (15.5 km—CPT) | 0.09 | 0.02 | −0.01 | 0.05 |
| Lower stratosphere (CPT—20 km) | −0.13 | −0.27 | −0.29 | −0.13 |

During the experimental period, the potential temperature profiles varied significantly in the UTLS (Figure 6a), particularly in the upper troposphere. The potential temperature lapse rate (unit: K/km) within three heights in the range 10–16 km was fitted using the first-order linear piecewise method (Table 4). The potential temperature lapse rates in the ranges of 10–11 km (Figure 6b) and 11–12.5 km (Figure 6c) corresponded to the minimum and maximum recorded values on 13 and 16 August 2018, respectively. Under the control of the high-intensity ASMA on 14 August 2018, the potential temperature lapse rate was approximately equivalent to the values recorded on 12 and 16 August. In particular, in the 10–11 km range, the potential temperature lapse rate on 14 August was approximately 2.2 times as high as that on 13 August owing to high-pressure activity at 500 hPa. This shift suppressed the vertical flow of the atmosphere and enhanced the static stability of the UTLS.

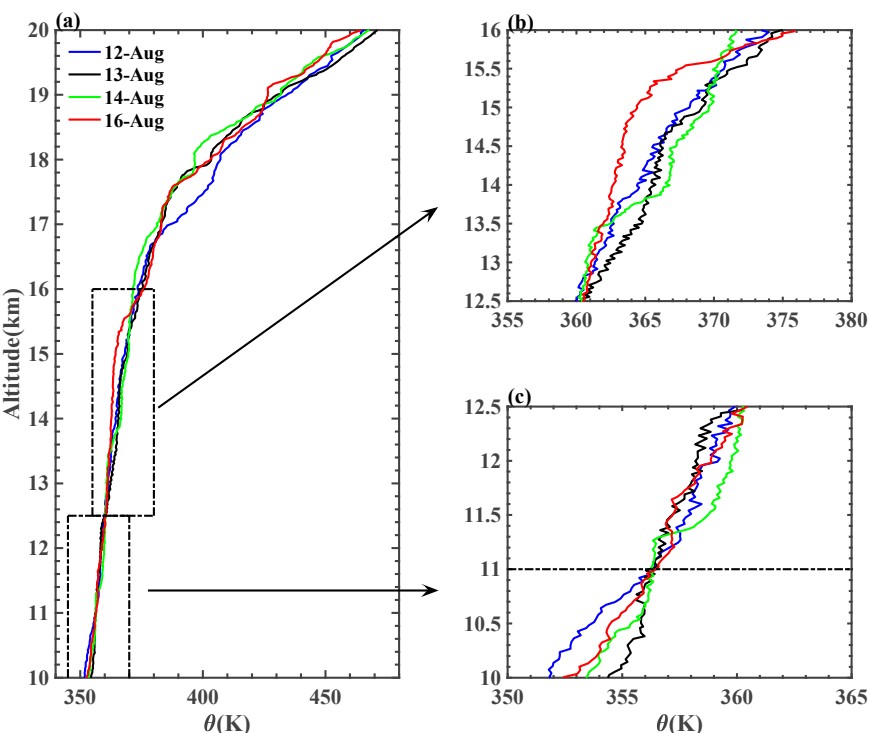

**Figure 6.** (**a**) Vertical profiles of potential temperature on 12–14 and 16 August 2018. (**b**,**c**) show magnified views of the 12.5–16 km and 10–12.5 km ranges, respectively.

**Table 4.** Potential temperature lapse rate (K/km) within the range of 10–16 km.

|  | **12 August 2018** | **13 August 2018** | **14 August 2018** | **16 August 2018** |
|---|---|---|---|---|
| 10–11 km | 4.84 | 1.48 | 3.29 | 3.48 |
| 11–12.5 km | 2.19 | 2.03 | 2.96 | 2.6 |
| 12.5–16 km | 3.79 | 3.65 | 3.73 | 3.24 |

In general, the potential temperature lapse rates for the four days were at the same amplitude within the range of 12.5–16 km. However, the trend of potential temperature on 16 August 2018 was significantly different from that on the other three days within 12.5–16 km. There was a weak thermal inversion layer in the range of 15–16 km on 16 August 2018. The inversion layer was able to block the upward movement of air [45,46], corresponding to a small $C_n^2$ value in the range of 10–15 km.

Piecewise linear fitting was performed on the potential temperature profiles within the tropopause and lower stratosphere regions at intervals of 1000 m. The CPT height of the TP was approximately 100 hPa, and the potential temperature lapse rate within 2 km below the CPT varied significantly among the four profiles. The potential temperature change rate on 14 August during the ASMA was twice as high as that on 12 and 16 August. This indicates that the stronger the ASMA, the greater the potential temperature lapse rate in tropopause. Above the CPT, $C_n^2$ cannot escape the fate of being affected by the ASMA, but thermal convection is inhibited in the lower stratosphere. The dynamic and thermal structures of the troposphere and stratosphere are completely different, and this difference is mainly characterized by high stability and weak turbulence in the stratosphere [14,47,48].

The tropopause is the mixed layer between the troposphere and stratosphere and it has dual characteristics of both the troposphere and stratosphere [49–51]. The CPT height corresponds to the minimum saturated water vapor mixing ratio, which is considered to be the upper boundary of the tropical tropopause [52]. Figure 7 shows that the potential temperature lapse rate is completely different between the tropopause and lower strato-sphere, mainly showing that the potential temperature lapse rate increases sharply in the

lower stratosphere. The turbulence characteristics of the lower stratosphere are hardly affected by the high-pressure activity over the 500-hPa layer, and the ASMA is the strongest influencing factor. Figure 2c shows that the Lhasa area was affected by the decreased ASMA on 12–14 and 16 August 2018. The potential temperature lapse rate in the tropopause area and above the CPT reached a maximum on 14 August and a minimum on 12 August 2018. The area 2 km lower than the CPT belongs to the tropopause mixed layer, and the potential temperature lapse rate in this area was largest on 14 August; the other three days did not differ extensively. It is inferred from the current data that the presence of the ASMA inhibits the vertical movement of the atmosphere in the lower stratosphere and tropopause. However, this conclusion requires additional data for verification.

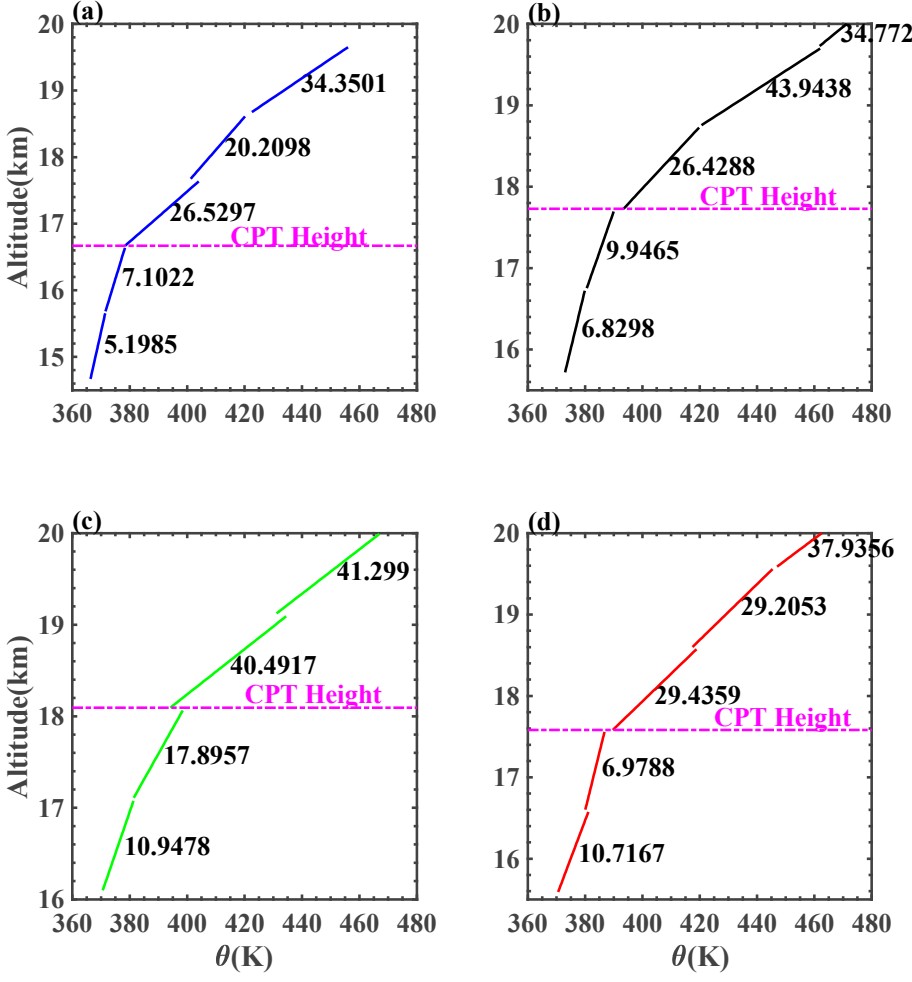

**Figure 7.** The piecewise fitting lines of potential temperature in tropopause and lower stratosphere. The potential temperature profiles are linearly fitted with the least-square method at intervals of 1000 m. The numbers indicate the coefficients (unit: K/km) of the piecewise fitting, indicating the potential temperature lapse rates. The pink dotted lines represent the corresponding CPT heights. (**a**) 12 August 2018, (**b**) 13 August 2018, (**c**) 14 August 2018, and (**d**) 16 August 2018.

## 5. Conclusions

In this study, the impacts of the ASMA and high-pressure activities in the 500 hPa layer on $C_n^2$ were analyzed under different ASMA strength states over the Lhasa during the summer based on precious in situ sounding data.

The atmospheric refractive index structure constant $C_n^2$ characterizes the optical turbulence intensity, which is directly affected by the atmospheric temperature. The ASMA is a warm, high-pressure system in the upper troposphere that causes the TP to be a strong heat source during the summer, heating the air over the TP. The upper atmosphere diverged to



form a high-pressure circulation system, and the lower atmosphere converged to form a low-pressure circulation system. The "upper highs and lower highs" pressure structure enhances the potential temperature lapse rate, which is conducive for the reduction in static stability and development of optical turbulence in the UTLS. However, once strong high-pressure activity exists in the lower pressure layer, such as at 500 hPa, the high-pressure system is dominant from the 500- to 100-hPa layer, constituting an "upper highs and lower highs" pressure field structure. In comparison with the "upper highs and lower lows" pressure field structure observed in most cases, this particular pressure field structure suppresses the vertical potential temperature lapse rate and vertical upward movement, and weakens the atmospheric convective activity. Under the combined action of the ASMA and low-pressure activity over 500 hPa, the potential temperature lapse rate decreased rapidly, and $C_n^2$ increased by an order of magnitude in the upper-troposphere.

The situations in both the tropopause and lower stratosphere are different from those in the upper troposphere, where atmosphere is almost unaffected by high-pressure activities at 500 hPa. The difference in the potential temperature lapse rate caused by the ASMA is particularly manifested in the region adjacent to the tropopause. The best evidence is that the potential temperature lapse rate in high-intensity ASMA situations is twice as high as that in low-intensity states. In other words, the potential temperature gradient can not only reflect the static structure of the atmosphere represented by buoyancy frequency, but also qualitatively analyze the variation tendency of $C_n^2$. Under different ASMA intensities, the potential temperature lapse rate is consistent with the variation tendency of $C_n^2$, and the profile on 13 August 2018, was the most evident.

The tropopause height over the TP is close to the 100-hPa layer, corresponding to the scope of activity of the ASMA. The tropopause height is closely related to the turbulence intensity. Strong turbulence elevates the CPT to a higher position, and the CPT temperature is lowered [42,53]. When the impact of the ASMA is greater, the CPT height rises by approximately 1.5 km.

In general, the ASMA has different mechanisms of influence on the atmospheric refractive index structure constant $C_n^2$ in the upper troposphere, tropopause, and lower stratosphere. It was found that during high-intensity ASMA, turbulent activity in the tropopause and lower stratosphere (in the upper-troposphere) is suppressed (promoted), which may be not conducive to the STE process (astronomical observations).

The extent of the promotion of convection by ASMA is not only related to the position of its center and strength but is also inseparable from the high-pressure activities of the lower atmosphere. Clearly, because of the limited radiosonde data considered in this study, determining whether our discussion is regional and limited and the reasons for the short-term $C_n^2$ fluctuations necessitate further exploration and analyses with various and abundant data, such as Stereo-SCIDAR measurements [26].

**Author Contributions:** K.Z.: methodology, software, data analysis, writing—original draft. F.W.: investigation. X.W.: data curation, project administration. N.W., X.L.: writing—review and editing. T.L.: project administration, funding acquisition, formal analysis, conceptualization, writing—review and editing. All authors have read and agreed to the published version of the manuscript.

**Funding:** This research was funded by the Strategic Priority Research Program of Chinese Academy of Sciences (Grant No. XDA17010104), and the National Natural Science Foundation of China (Grant Nos. 4157685 and 91752103).

**Data Availability Statement:** In this section, the data underlying this article cannot be shared publicly due to the confidentiality requirements of the project in the study.

**Acknowledgments:** Thanks to the reanalysis data provided by the European Centre for Medium-Range Weather Forecasts (ECMWF) for the purposes of this study.

**Conflicts of Interest:** The authors declare no conflict of interest.

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
