# Peer review of "Optical Turbulence Characteristics in the Upper Troposphere–Lower Stratosphere over the Lhasa within the Asian Summer Monsoon Anticyclone"

_remotesensing, doi:10.3390/rs14164104_

Round 1
Reviewer 1 Report
The paper analyzes characteristics of the atmospheric refractive index structure constant (Cn2) profiles in the Lhasa area at different strength states of the Asian summer monsoon anticyclone (ASMA) based on precious in-situ sounding data measured over Lhasa. I would suggest this paper to be accepted after minor revision. Comments are given below.
1. The used thermal turbulence sounding data is quite limited in this paper. Considering that the behavior of Cn2 is closely related to the meteorological, analysis of meteorological parameters form other sources can benefit the discussions of this paper, such as the analysis of temperature structure during ASMA in section 4.2. So I suggest that the authors added some results from conventional meteorological sounding data from meteorological bureau or reanalysis data in section 4.2.
2. This paper mainly analyzes Cn2, except giving a parameterization of Ct2 in section 4.1. I would suggest that the authors rewrite this section into form of Cn2.
3. In section 4.2, Figure 4 shows results from 12 to 16 August, including 15 August. However, results of 15 August are not shown in the rest part.
Author Response
Dear reviewer:
Thank you for your decisions and constructive comments on our manuscript. We have carefully considered the suggestion of Reviewer and make some changes. We have tried our best to improve and made some changes in the manuscript. Accordingly, we have uploaded a copy of the original manuscript with all the changes highlighted by using the track changes mode in MS Word. Revision notes, point-to-point, are given as follows:
Reviewer 1:
The paper analyzes characteristics of the atmospheric refractive index structure constant (Cn2) profiles in the Lhasa area at different strength states of the Asian summer monsoon anticyclone (ASMA) based on precious in-situ sounding data measured over Lhasa. I would suggest this paper to be accepted after minor revision. Comments are given below.
1. The used thermal turbulence sounding data is quite limited in this paper. Considering that the behavior of Cn2 is closely related to the meteorological, analysis of meteorological parameters form other sources can benefit the discussions of this paper, such as the analysis of temperature structure during ASMA in section 4.2. So I suggest that the authors added some results from conventional meteorological sounding data from meteorological bureau or reanalysis data in section 4.2.
Reply: According to Reviewer 2's comment, because the research area of the study is Lhasa site, the title has been changed to “Optical turbulence characteristics in the upper troposphere–lower stratosphere over the Lhasa within the Asian summer monsoon anticyclone”.
Because of the low vertical resolution of the reanalysis data, it is difficult to capture the fine structure of the atmosphere in the UTLS. It is also very difficult to obtain the conventional meteorological sounding data of the local weather bureau in a short period of time.
Turbulence data over the TP is inherently difficult to obtain, and it is even more difficult and rarer to capture an ASMA activity during the experiment, which in return results in a relatively limited data sample size in this study.
This study is a preliminary research on “the distribution characteristics of the vertical atmospheric turbulence structure in UTLS over the TP and its influencing factors”. Atmospheric turbulence sounding experiments at other stations on the TP are also being planned, and the turbulence data on the TP is also being collected step by step for deeper mechanism exploration.
2. This paper mainly analyzes Cn2, except giving. I would suggest that the authors rewrite this section into form of Cn2.
Reply: Thank you for the suggestion. At the beginning of Section 4.1, a transitional sentence was added. “Eq.(2) indicates that the most critical step in the parameterization scheme of atmospheric turbulence is how to parameterize CT2, and Cn2 can be calculated logically.” (Line 301). Thus, the parameterization of Ct2 in section 4.1 can introduced from the transitional sentence.
3. In section 4.2, Figure 4 shows results from 12 to 16 August, including 15 August. However, results of 15 August are not shown in the rest part.
Reply: As described in Table 1, due to the storm, micro-thermal sensor was destroyed, resulting in the lack of turbulence data on the August 15, 2018. Because there is no comparison of turbulent data, Section 4.2 does not describe the conventional weather conditions for the day.

Reviewer 2 Report
The authors presented the manuscript "Optical turbulence characteristics in the upper troposphere– 2 lower stratosphere over the Tibetan Plateau under different 3 Asian summer monsoon anticyclone strength states" which contains the results of studies of the structure of optical turbulence over the plateau over the Tibetan Plateau.
The study is original and important from the point of view of applications, namely, the placement within the Tibetan Plateau of new high-resolution ground-based astronomical telescopes that need detailed information about the structure of the atmosphere.
In the manuscript we note a number of comments.
- Firstly, the authors set the goal of studying turbulence over a fairly large area (over the Tibetan Plateau). We ask you to align the presented results over Lhasa or justify your approach (equivalence of the results over Lhasa with the results over the extensive over Tibetan Plateau)
- Secondly, why optical turbulence profiles above 10 km were chosen, whereas the greatest optical turbulence is observed in the lower atmospheric layers. You use a standard gradient method for modeling optical turbulence, i.e. you do not offer new methods. Justify the need to determine the optical turbulence profiles over Lhasa above 10 km. What is the novelty and significance of the results?
-We recommend the authors to give some brief overview of methods for determining the characteristics of optical turbulence (profiles) in the introduction. We recommend the following works to the authors:
1. Kovadlo, P.G. et al. Study of the Optical Atmospheric Distortions using Wavefront Sensor Data. Russ Phys J 63, 1952–1958 (2021). https://doi.org/10.1007/s11182-021-02256-y
2. Butterley, T.; Wilson, R.; Sarazin, M. Determination of the profile of atmospheric optical turbulence strength from SLODAR data. Mon. Not. R. Astron. Soc. 2006, 369, 835–845
3. Shepherd, H.W.; Osborn, J.; Wilson, R.W.; Butterley, T.; Avila, R.; Dhillon, V.S.; Morris, T.J. Stereo-SCIDAR: Optical turbulence profiling with high sensitivity using a modified SCIDAR instrument. MNRAS 2014, 437, 3568–3577.
4.Odintsov, S.L.; Gladkikh, V.A.; Kamardin, A.P.; Nevzorova, I.V. Determination of the Structural Characteristic of the Refractive Index of Optical Waves in the Atmospheric Boundary Layer with Remote Acoustic Sounding Facilities. Atmosphere 2019, 10, 711
- Also , vertical profiles of optical turbulence in terms of CN2 are defined for a limited period of time from August 12 to August 16 , 2018 . We propose to indicate to the authors what features these profiles have in comparison with the profiles of CN2 values averaged over long time intervals (these profiles at high altitudes have an order of magnitude smaller CN2 values) We recommend using the results of the work:
5. Y. Hach, A. Jabiri, A. Ziad, A. Bounhir, M. Sabil, A. Abahamid and Z. Benkhaldoun Meteorological profiles and optical turbulence in the free atmosphere with NCEP/NCAR data at Oukaımeden – I. Meteorological parameters analysis and tropospheric wind regimes / Mon. Not. R. Astron. Soc. 420, 637–650 (2012)
6. J Osborn et al Optical turbulence profiling with Stereo-SCIDAR for VLT and ELT // Monthly Notices of the Royal Astronomical Society, Volume 478, Issue 1, July 2018, Pages 825–834, https://doi.org/10.1093/mnras/sty1070
- We would like to note the absence of a maximum at an altitude of 200 hPa. We recommend the authors to explain the absence of this maximum from a physical point of view.
Author Response
Dear reviewer:
Thank you for your decisions and constructive comments on our manuscript. We have carefully considered the suggestion of Reviewer and make some changes. We have tried our best to improve and made some changes in the manuscript. Accordingly, we have uploaded a copy of the original manuscript with all the changes highlighted by using the track changes mode in MS Word. Revision notes, point-to-point, are given in the attached file.

Reviewer 3 Report
--see attached

Author Response
Dear reviewer:
Thank you for your decisions and constructive comments on our manuscript. We have carefully considered the suggestion of Reviewer and make some changes. We have tried our best to improve and made some changes in the manuscript. Accordingly, we have uploaded a copy of the original manuscript with all the changes highlighted by using the track changes mode in MS Word. Revision notes, point-to-point, are given as follows:
Reviewer 3:
The study presents the use of radiosonde observations to analyze the optical turbulence variations over Tibetan Plateau through atmospheric refractive index structure constant during atmospheric refractive index structure constant activity of various strength.
Overall, a well written manuscript with some minor issues. Very few details were provided regarding Cn2 and its significance or connection to the optical turbulence. Section 2 could be improved in this regard. The figures were fine and legible. I recommend publication after addressing the following minor comments.
Reply: Thank you for the suggestion. We have added the information required as explained above. “the degree of refractive index fluctuation due to variations in atmospheric temperature and density” (Line 130) and the corresponding references ([18, 32]) were cited in the text.
Minor comments:
Comment #1: Apart from Figure 1, the ECMWF data was not used or discussed elsewhere in the paper. I believe you don’t need a separate section (Sec 2.2) for that, you can simply include those details near the discussion around Figure 1.
Reply: According to comment, the content in Section 2.2 has been deleted and moved it into the caption of Figure 2 concisely.
Comment #2: Clearly define Cn2 and Cn2 at their first occurrence. At Lines 82 & 102, Cn2 is mentioned as temperature structure constant. I assume this is a typo and authors meant to use Cn2 for temperature structure constant, as written in Sec 4.1.
Reply: Thank you for the suggestion. We have modified this expression throughout the text according to the comment (Line 98, Line 246 and Line 453).
Comment #3: Authors may include elevation map of TP or study region as part of Figure 1.
Reply: Thank you for the suggestion. To maintain image consistency and aesthetics, we have added the elevation map of TP and the geographical location of Lhasa (Figure 1) required as explained above. The serial numbers of other pictures are sequentially followed.
Comment #4: Line 147: Do you mean color circle?
Reply: Thank you for underlining this deficiency. “Color circle” rewritten as “dots” (Line 202), referring the blue, black, green, grey and red dots in Figure 2(a).
Comment #5: You don’t need additional axis in Figure 2(d), as both measure geopotential anomaly. Instead, use legend. Also, try keeping the same date range for Figure 2(b)- (d).
Reply: Thank you for the suggestion. The abscissa of Figures 2c and 2d were adjusted to be the same as that of Figure 2b, from August 12 to August 20, 2018.
Comment #6: Line 194: “... anomaly...” to “... geopotential anomaly...’
Reply: Thank you for the suggestion. This sentence was rephrased according to the comment (Line 269: ... geopotential anomaly...).
Comment #7: Line 197: “...enhanced the static stability...”, is any analysis done to know this? If so, presenting them as a plot or giving some values here would benefit the reader.
Reply: Thank you for the comment. The sentence is really inappropriate for appearing in its original place. We have moved the sentence to Line 359 according to the comment. The potential temperature gradient summarized in Table 4.
Comment #8: Line 308: I think it is meant to be “... upper highs and lower lows...”.
Reply: Thank you for the suggestion. We have modified eight similar expressions throughout the text according to the comment (Line 18, Line 22, Line 265, Line 270, Line 422, Line 431 and Line 432).

Round 2
Reviewer 2 Report
Authors made changes according to our comments. The manuscript may be recommended for publication.